# Mapping the Field in Stress, Anxiety, and Postpartum Depression in Mothers of Preterm Infants in Neonatal Intensive Care

**DOI:** 10.3390/children8090730

**Published:** 2021-08-25

**Authors:** Delia Cristóbal-Cañadas, Antonio Bonillo-Perales, María del Pilar Casado-Belmonte, Rafael Galera-Martínez, Tesifón Parrón-Carreño

**Affiliations:** 1Pediatric, Intensive Care Unit, Torrecárdenas University Hospital, 04009 Almería, Spain; deliacristobal@gmail.com; 2Pediatric Department, Torrecárdenas University Hospital, 04009 Almería, Spain; abonillop@gmail.com (A.B.-P.); galeramartinez@gmail.com (R.G.-M.); 3Department of Economics and Business, University of Almería, 04120 Almería, Spain; 4Department of Nursing, Physiotherapy and Medicine, University of Almería, 04120 Almería, Spain; tpc468@ual.es

**Keywords:** stress, anxiety, postpartum depression, preterm infants, state of the art, bibliometric

## Abstract

Objective: The main aim of this study was to describe and conduct a bibliometric analysis of the state of research on stress, anxiety, and postpartum depression in mothers of preterm infants in the Neonatal Intensive Care Unit. Background: Women affected by premature births are particularly exposed to mental health difficulties in the postpartum period. The desire to comprehend and the growing interest in research on stress, anxiety, and postpartum depression in mothers of preterm infants in neonatal intensive care have led to a substantial rise in the number of documents in this field over the last years. Thus, it makes it vital to regularly review the state of knowledge on this phenomenon in order to identify progress and constraints, to stimulate reflection, and to encourage progress in future research. Method: This study examined 366 articles published in the Scopus database (1976–2020). Keyword analysis was also used to identify hot research trends to be developed in future studies. This study complies with the PRISMA-Scr guidelines for quality improvement research in the EQUATOR network. Results: Our results reveal that research in this field is in a period of high production and allows this flourishing body of work to be organized into different periods, highlighting the most important themes. In such a way, our research enriches the lively field by presenting a comprehensive understanding of the field. Discussion: The key contribution of this study is the development of a conceptual map of research on stress, anxiety, and postpartum depression in mothers of preterm infants in neonatal intensive care units.

## 1. Introduction

According to the World Health Organization (WHO) report [1], almost 15 million children are born prematurely each year, which is one-tenth of all newborns, and the statistics for these children vary from 5% to 18% in different countries. Stress, anxiety, and even depression can reflect a range of cognitive and behavioral efforts to cope with stressful life events. Parents of preterm infants who are admitted at a neonatal intensive care unit (NICU) experience increased stress because they are rarely prepared for the shock, stress, and anxiety when their child needs critical care in a NICU [2,3,4,5]. According to the WHO [1], premature birth (PTB) is a major cause of infant mortality and morbidity. Children born prematurely (before 37 completed weeks of gestation) have a higher risk of diseases than children born on due date. Premature infants are at increased risk of several health and developmental problems, and they have significant emotional and economic costs for families and significant consequences for public services.

Despite decades of research, the prevalence of preterm births has not decreased, and their etiology remains unexplored. Research has shown that preterm birth is a worrying event for parents, who often report stress and anxiety [6,7,8]. Parenting in the NICU can cause great psychological anxiety [9]. The parent–child relationship begins before birth and develops later; however, if the birth occurs earlier than expected, or even too early, it can interfere with the normal bonding process. Although both parents are undoubtedly influenced by the NICU experience, mothers may suffer a greater impact. Many women may experience a wide range of psychiatric symptoms with mood disorders and perinatal anxiety even in the context of a normal birth, which are among the most common mental health effects [10]. Several studies suggest that stressful reproductive events, such as admission to a neonatal intensive care unit, may provoke or exacerbate the onset of pre-existing maternal psychiatric symptoms, and postpartum depression (PPD) rates were found to be significantly higher in mothers with NICU than in the general population [11,12]. PPD is a common and serious mental health problem, which is related to maternal stress and anxiety and different negative consequences for infants [13]. Regarding anxiety and stress, it is well documented that postnatal anxiety and adjustment disorders are associated with variables specific to NICU, such as very low birth weight, preterm birth, perinatal complications, and stressful birth experiences [14,15]. It has been shown that a prolonged stay in hospital can have a destructive effect on the bonding process [16]. Although there have been changes in the NICU in recent decades that have made it easier for parents to be present while their child is in hospital, the NICU remains a stressful environment for parents, as many studies have shown [17,18].

The physical environment of the NICU is characterized by monitoring equipment, wires, and tubes that are connected to children. Recent studies show that maternal functions and mother–infant interactions in mothers of preterm infants in the NICU may be impaired [19,20]. However, the greatest stress experienced by parents is associated with separation from the child and loss of the parental role they had previously imagined, associated with a lack of physical and emotional closeness, which is an important factor in the early relationship between parents and the newborn [21]. In fact, as it is often reported in the literature, the first moments of the postnatal phase are crucial in establishing an early parent–child relationship [22]. While the child is in hospital, mothers may experience different, often contradictory emotional reactions such as loss of self-esteem, pain, feelings of failure, sadness, guilt, fear, and anger [23]. The literature shows that early symptoms of depression in mothers have negative effects on their relationship with their child and on their parental role, especially after preterm delivery [24]. In recent times, PPD has received increased attention as it has been recognized as one of the most common morbidities in the perinatal period [13]. In addition to its prevalence, it is estimated at about 14% [25] that PPD is considered to be a devastating morbidity because it affects more than just the mother’s health [26], as it could also affect the child’s development [27,28,29]. PPD disrupts the mother–child bond, limiting a woman’s ability to fulfil function in the maternal role and increasing the likelihood of early cessation of breastfeeding, as they often have problems with breastfeeding [30,31].

Although most research on the influence of the mother’s postpartum mood on the child’s development focuses primarily on PPD, researchers have found in recent decades that postpartum anxiety has independent effects, as do stress and PPD [32,33]. Since the 1980s, a growing number of theoretical documents have appeared with a focus on the analysis of the psychological processes faced by mothers of preterm infants admitted to NICU [34]. In such a way, the stress, anxiety, and depression of these mothers have gained attention from the research community. The vast literature regarding this topic needs to be organized, establishing the cognitive and social structure of the research field. To the best of the authors’ knowledge, there are no studies using bibliometric techniques regarding stress, anxiety, and PPD in mothers of preterm infants in NICU.

Bibliometric analysis is important to elucidate the most important factors and best-researched topic areas in this field. In the field of psychological factors affecting mothers of preterm infants admitted to intensive care units, it has been suggested that bibliometric studies can improve our understanding by analyzing their evolution and intellectual core. Bibliometric analysis is apt, given that, in all fields, in both practical and theoretical research, it is sometimes necessary to delve into past research and determine the direction of future research [35]. Moreover, the use of bibliometrics guides not only experts in the field, but also researchers who want to start their investigations in the field [36].

Therefore, the purpose of this analysis is to present the current situation of research on the stress, anxiety, and PPD in mothers of preterm infants in NICU, based on studies published between 1976 and 2020 in the Scopus database, in order to establish the social networks of the field and the cognitive structure of the field. In such a way, the main trends of research are identified, and future research lines are detected.

Through the bibliometric evaluation of 366 articles, this study contributes to research on stress, anxiety, and PPD in mothers of preterm infants in NICU in several ways. As a first contribution, it provides a full overview of this field of research over the period 1976–2020, detailing performance indicators such as annual published documents, articles, and authors with the major number of citations, as well as journals publishing the greatest number of articles in the field. In addition, it shows the network of relationships among authors. Lastly, as a third contribution, it includes a strategic framework, based on a joint analysis of keywords, to discover the most researched and most relevant topics. This literature review is, therefore, an exceptionally comprehensive study which extends existing research on the subject by extending the analysis period and refocusing on the area.

## 2. Methodology

This study adhered to the PRISMA-ScR guidelines for quality improvement as part of the EQUATOR initiative (Enhancing the Quality and Transparency of Health Research, 2018; www.equator-network.org; accessed on 1 February 2021). Earlier articles [37,38] have suggested that bibliometric analysis should follow five steps: (1) defining the research field, (2) selecting the database, (3) adaptation of the search criteria, (4) codifying the retrieved material, and (5) verification of the information. The current study follows these steps for a transparent and iterative review process. A full overview of this process can be found in Figure 1.

### 2.1. Definition of the Field of Study

As specified in the introduction section, a comprehensive bibliometric analysis of research on stress, anxiety, and PPD in mothers of preterm infants in NICU is vital so as to organize the research field. Therefore, the main purpose of this study was to offer performance indicators of the scientific production in the field, as well as show the social and intellectual networks in the research field.

### 2.2. Database Selection

Traditionally, bibliometric studies have been conducted on the basis of the Web of Science (WoS) database [38,39]. Nevertheless, Scopus appeared in 2004 as an alternative database, competing directly with WoS [39]. In this vein, Scopus and WoS are thought to be the two major databases for conducting bibliometric analyses. The present study uses the Scopus database due to several reasons. First, it was discovered that 84% of manuscripts that are indices in WoS were also included in Scopus [40]. Second, Scopus covers a wider number of journals than WoS, reducing in such a way the risk of omitting important manuscripts during the search.

### 2.3. Adjustment of Research Criteria

The definition of a proper search criteria remains vital in bibliometric analyses so as to retrieve suitable information [35]. Accordingly, to achieve a wide range of word combinations, the following parameters were applied to the search: “Title-Abstract-Keyword” (“stress” OR “psychological stress” OR “anxiety” OR “postpartum depression*” OR “postpartum-depression*” OR “postnatal depression*” OR “post-natal depression*”) AND (“mother*”) AND (“neonatal intensive care”) AND (“premature*” OR “preterm*” OR “premature infant”). The period of time was restricted to 1976–2020, since the first manuscript on this subject was found in the Scopus database in 1976. In addition, the search was limited to articles, excluding review articles, books, book chapters, and conference papers. The final number of manuscripts was made up of 366 articles.

### 2.4. Codification of Recovered Material

Data were extracted from the Scopus database by downloading ris and csv format files. The software used to analyze the data was Excel, VOSviewer (v1.6.9), and SciMAT (v1.1.04). Fist, output indicators (e.g., published papers) and performance indicators (e.g., h-index) were obtained to create tables, illustrate descriptive graphs, and code data using Excel. Second, bibliometric software was used to identify the social network (VOSviewer) and the intellectual network (SciMAT).

### 2.5. Examination of the Information

As shown at the coding stage of the retrieved material, some data (calculations and definitions of productivity indicators) were validated using Excel (e.g., formula calculations) and SciMAT (e.g., validation procedures). Two complementary bibliometric tools were used; VOSviewer and SciMAT were used to describe and identify scientific maps [41]. VOSviewer is the most widely used bibliographic software and aids to identify collaborative networks between countries and authors [41]. SciMAT, on the other hand, is a very useful software tool that allows you to identify emerging research trends in a given research area by analyzing a set of words together [42].

The results of the co-word analysis are shown in SciMAT using a strategy diagram divided into four quadrants (see step 4 in Figure 1) [43]. The research themes and thematic networks discovered are visualized. The themes located in the upper right quadrant are deemed to be motor topics because of their strong centrality and high density. Centrality and density are the two indicators that characterize each theme. The degree of interaction with other networks is measured by centrality and the internal strength of the network is measured by density. These topics are well advanced and relevant to the cognitive core of the field. In the lower right quadrant, we find the general themes that affect multiple areas. These themes are relevant but not yet developed. In the lower left quadrant are themes that are thought to be developing or declining due to their low density and low centrality. Lastly, peripheral to the domain are those issues that are in the upper left quadrant, i.e., internal linkages are well developed, and external linkages are insignificant. Furthermore, the strategy diagram contains a third dimension. Spheres (representing issues) appear in the quadrant, and the size of each one depicts the number of manuscripts in which the keyword exists [42].

## 3. Results and Discussion

The present bibliometric analysis contained a total of 366 articles from 160 authors belonging to institutions in 46 countries and published in 160 journals, citing 10,118 references.

### 3.1. Descriptive Analysis

Table 1 (Appendix A) shows that the number of articles published on this field has gradually increased since 1976. The temporal evolution indicates three stages of the trend in the number of publications. The first covers the period from 1976 to 2002, when few publications were published. In the following years, from 2003 to 2011, the number of documents published increased, with 5.9 being the average number of manuscripts published per year. From 2012, more attention was given to research into stress, anxiety, and PPD in mothers of preterm infants in NICU, highlighting 2019 as the year in which most publications were published. Although there was a decline in research in some years, the mean of manuscripts published annually was 29, indicating that 72.91% of research on the subject has been published over the past 8 years, and this trend is increasing. Figure 1 also shows that the number of citations and published articles has increased similarly, indicating the low productivity of early years, common to other studies [44].

### 3.2. Distribution of Scientific Production

According to the most important thematic areas in which the scientific production on stress, anxiety, and postpartum depression in mothers of premature babies in the NICU is classified by Scopus, the most relevant thematic area is Medicine with 54.9%, followed by Nursing with 21.8%. Two other relevant thematic areas are Psychology with 11.3% and Social Sciences with 2.6%. These four disciplines account for 90.26% of the total number of published articles, while another eight fields together account for the remaining 9.4% of published articles (Appendix A).

Research on stress, anxiety, and postpartum depression in mothers of preterm infants in NICU was published in 366 different Scopus journals between 1976 and 2020. The journal Advances in Neonatal Care and Human Development lead with 17 articles, followed by the Journal of Perinatology with 10 articles. In addition, the data show that 98.12% of the other journals published between one and nine articles (Appendix A). The journal with the most citations (C) was Early Human Development with 617 and the Journal of Perinatology with 533. When considering the average citations per article (C/A), the Pediatrics journal appeared at the top with 143.1 citations per article, followed by the Journal of Perinatology with 53.30 and the Early Human Development Journal with 36.29 citations per article. Since the year of publication could have an effect on the study of the influence, the average number of citations per articles since the year of the first published article (C/Y) was obtained so as to reduce the effect. According to the C/Y indicator, Early Human Development led with 36.29 citations per year, while the Pediatrics journal with 34.69 reached second place. With regard to the h-Index, there were three major journals: Early Human Development, Advances in Neonatal Care, and the Journal of Perinatology, all with an h-Index equal to or greater than 9. Lastly, another aspect worth noting is that 90% of the top 10 journals belong to Europe and the United States. Specifically, four are from the United States, three are from the United Kingdom, one is from the Netherlands, and one is from Denmark, revealing that journals from these regions are at the vanguard of stress, anxiety, and postpartum depression research in mothers of preterm infants in NICU.

### 3.3. Authors and Papers

This study identified the 10 best and most influential authors in the ranking (Appendix A). These 10 authors are affiliated with nine institutions, from six different countries. In particular, two of these authors are affiliated with Scientific Institute IRCCS Eugenio Medea (Italy). The authors of the most frequently published articles are Holditch-Davis, D. with nine, followed by Miles, M.S. with seven articles and Falacking, R., Montirosso, R., and Shaw, R.J. with six articles each, from Norway, Italy, and the United States respectively. Miles, M.S. is the author with the most citations (C), followed by Holditch-Davis, D. In terms of citation per author, the most quoted is Miles, M.S. followed by Holditch-Davis, D. Other important authors are Horwitz, S.M. with five articles and 224 citations and Mörelius, E. with five articles and 198 citations. Lastly, it should be mentioned that most of these authors have more than 6 years of publishing experience on this topic.

Figure 2 depicts a network of relationships among authors who have at least three documents in common. Two different groups can be observed. The first (blue) is led by Anderson, P.J. As mentioned above, he is one of the leading authors of research on anxiety and postpartum depression and a relevant author in the field of the impact of maternal postpartum depression on child development, as well as the behavior and development of premature infants. He works primarily with Doyle L.W. (University of Melbourne, Melbourne, Australia), Gemmill, A.W., (Parent–Infant Research Institute, Melbourne, Australia), Hunt R.W. (Murdoch Children’s Research Institute, Melbourne, Australia), Milgrom J. (University of Melbourne, Melbourne, Australia), and Newnham C. (University of Melbourne, Melbourne, Australia). The second group (green) is led by Inder, T. (University of California, CA, USA) and Pineda R.G. (Washington University, DC, USA).

The three most frequently cited articles from the period 1976–2020, all of which represent the works with the greatest influence on the field of research on stress, anxiety, and postpartum depression in mothers of preterm infants in NICU, are the following (Appendix A): the most frequently cited article is “*Maternal psychological distress and parenting stress after the birth of a very low-birth-weight infant*” by Singer L.T., et al. [45], which contains 528 citations. This empirical work shows the degree and nature of stress experienced over time by mothers whose children differ in terms of the degree of premature birth and medical and developmental risks. Als et al. [46] were the authors of the second most cited article “*Individualized behavioral and environmental care for the very low birth weight preterm infant at high risk for bronchopulmonary dysplasia: Neonatal intensive care unit and developmental outcome*”, accounting for 376 citations. In this article, they developed a behavioral observation method for premature newborns that catalogs specific reaction patterns according to assumed stress and relaxation behavior. The third most cited article is “*Reducing premature infants ‘length of stay and improving parents’ mental health outcomes with the Creating Opportunities for Parent Empowerment (COPE) Neonatal Intensive Care Unit Program: A randomized, controlled trial*” by Melnyk et al. [47]. This empirical work evaluated the effectiveness of the educational program of behavioral intervention, aimed at improving parent–child interaction and the mental health of parents, with the ultimate aim of improving children’s development and behavioral outcomes. To avoid the effect of the years of publication, the variable C/Y was obtained. As can be seen, the work with the greatest impact was the study of Melnyk et al., with 26.8 citations per year, but the second most impactful article was again by Singer et al.

LATLY, it is worth mentioning that two of the five best articles were published by the same journal: *Pediatrics*. According to the data, this journal ranks sixth in the number of research articles published.

### 3.4. Content Analysis

The study of research trends was divided into three subperiods, selected on the basis of criteria from Cobo [43], who recommends including an appropriate number of articles in each period, albeit using a different number of years. Thus, the first subperiod covered the years 1976 to 2002 and included a total of 39 articles. The second subperiod covered the years 2003–2011 and contained 59 articles. lastly, the third subperiod ran from 2012 to 2020 and comprised a total of 268 articles.

The investigation of the first subperiod (1976–2002) is shown in the strategic diagram in Figure 3. The four quadrants represent topics in the literature depending on their density and centrality. As far as the upper right quadrant is concerned, three movement topics are presented that have been carefully studied and are of crucial importance in this area, based on their high density and strong centrality. In this first subperiod, the motor topics were human, adult, infant, attitude to health, and mother–child relation. The position of human, adult, and infant does not come as a surprise since the ability of parents to build lasting relationships with their children is one of the fundamental characteristics of the human experience. Attachment is seen as a central force in the development of life. Women develop a bond with their infant during pregnancy, which lasts and develops more strongly after the infant is born [48]. These mothers are in particular upset by the separation and their inability to care for her premature child, which changes the development of the mother–child relationship [49].

As for the basic themes, it is those in the lower right quadrant that are considered basic, general, and cross-cutting themes. Two themes seem to dominate this quadrant: parents and adolescent. It is well documented that the birth and hospitalization of a premature infant is very distressing for the parents. Younger mothers seem to be more exposed to postnatal anxiety, the highest degree of general mental stress and probably less psychological copying capacity in the face of a high-risk birth and intensive hospitalization [50].

The lower left quadrant shows the emerging or disappearing themes. Figure 3 (Appendix A) highlights pregnancy, gestational age, and very low birthweight as topics with low density and centrality. Pregnancy, gestational age, and very low birthweight appear as themes. Very-low-birthweight delivery has become increasingly common due to medical advances in neonatal intensive care and risk pregnancy management, as well as the study of different maternal factors associated with admission of the preterm infant to neonatal intensive care [45].

The upper left quadrant indicates issues that are marginal or irrelevant in the field, albeit well-developed. These are child care and interview, and it is important to analyze their evolution in subsequent subperiods.

The second subperiod (Figure 4, Appendix A) included the years from 2003 to 2011 and revealed seven motor themes: female, intensive care neonatal, child–parent relation, hospitalization, randomized controlled trial, young adult, and child behavior. It deserves attention that the mother–child relation was in the first subperiod as an emergent topic and, in this second subperiod, the child–parent relation emerged as emergent. Most of the research into the parental experience after premature birth focused on mothers [51]; however, in recent years, more attention has been paid to the shared experience of mothers and fathers [47,52].

The literature has paid attention to the hospitalization of preterm infants in NICUs, and the characteristics of these units already influence the psychological wellbeing of the mothers, the child–parent relation, and the behavior of the newborn [53]. Fathers must cope with the emotional changes associated with parenthood and must play a supportive role toward their partner addressing these difficulties by providing protection from excessive mental suffering and promoting mother–child interaction.

The basic themes depict topics that are well developed and relevant. It is worth noting the initiative of some authors in this subperiod as a basic theme to further explore the profile of the psychological aspects. Results of multiple studies have documented that psychological aspects, such as parental emotional distress during the child’s hospital stay in the NICU and the experience of negative feelings, stress, and traumatic symptoms, combined with the premature infant’s physical and neurological immaturity, can change the parents role and the child’s early development [54,55] and can be an obstacle to building an effective and functional mother–child interaction [51]. Thus, it is a topic of interest because of the important influence on the infant and the long-term consequences. Preventive interventions can improve the experience of the ICU parent and can benefit these infants.

Lastly, anxiety, which was connected with pregnancy and premature infant, was shown as an emergent or decadent topics. Furthermore, there were other isolated topics such as nursing.

The analysis of the third subperiod (Figure 5, Appendix A) presented the following as driving themes: human, father, prospective studies, psychological wellbeing, and very low birthweight. Most research on the parent experience after preterm birth has centered on mothers, although more attention has recently been paid to the joint experience of fathers. Several studies have shown that the mother and father were emotionally affected, and that the experiences and contacts of both parents can impact the neurological development of preterm and very-low-birthweight infants, while also improving their psychological wellbeing; thus, so this topic has become a well-developed field of research [56,57]. However, studies on parents providing care to preterm or very-low-birthweight infants remain limited; therefore, the effects of intensive care interventions, such as skin-to-skin contact, on parents have not been well studied [58] and studies from a prospective perspective would be important. Furthermore, maternal mental health is associated with child developmental performance, which highlights the need to address maternal psychological symptoms as early as possible, promoting infant wellbeing by improving mother–child contact and maternal psychological wellbeing [59].

In this subperiod, preterm infants and preterm birth remained basic themes and were a broad topic of research. Preterm birth appears to affect mothers’ psychological state and their interaction with their baby. The clinical conditions of very preterm infants, as well as the mortality rate among these infants, may cause parents to experience an emotional state characterized by feelings of anxiety, great distress, and despair that may prevent mothers from relating to their baby. These symptoms can affect the mother’s emotions and determine her reactions, which are crucial for their interactions and the development of the mother–infant relationship. These emotions, such as depression or post-traumatic stress, are present at higher levels than in term newborns [60].

In the lower left quadrant, attitudes, vital signs, and mobile applications appeared as emerging or declining themes. Parental engagement in the care of preterm infants leads to superior breastfeeding rates, earlier secretion, and better long-term neurological progress [61]. Most studies have increasingly focused on parental attitudes and family care as a relevant and key component of NICUs; therefore, family care and family protection may be vital to identify worries and pressures associated with parental roles [62]. In addition, research has highlighted the importance of certain interventions, such as the kangaroo method, involving different positions of premature infants, as comfort measures that reduce stress by improving the stability of vital signs [63]. Similarly, it is considered that nursing care could be spurred through the use of electronic health records, such as mobile apps, as it could be helpful to reduce maternal stress [64].

Early stressful experiences may influence the long-term outcomes of preterm infants. Therefore, it seems likely that interventions aimed at reducing stress in the mother–infant relationship with a family-centered approach will continue to be an important part of research and practice aimed at optimizing parental wellbeing and the developmental and behavioral prospects of preterm infants in the near future [65,66].

### 3.5. Interpretation of the Analysis

So far, no detailed map of the research architecture on stress, anxiety, and PPD in mothers of preterm infants in the NICU has been established worldwide, covering aspects of research activity, as well as the dynamics and trend patterns of literature production, and identifying the types of documents and the most prolific authors. According to our results and in an attempt to make a comparison with previous similar studies on this topic, some similarities and differences were identified. First, they differed in terms of the period analyzed, the choice of database, and the search parameters. Brüggmann et al. [44] analyzed the period 1900–2012 in their research on maternal depression using Web of Science and ScienceDirect databases, and the sample consisted of 7330 articles; it should be noted that the search parameter used in this study was broader than that used in our research. Secondly, they agreed with this study, stating the United States as the most productive country. Thirdly, our findings point to similar conclusions on important or key issues. In this sense, these two studies on international development overlap as an issue that deserves more research in the future. As the authors stressed, despite the diversity of research on postnatal depression, there is a need to understand the impact of postnatal depression on newborns.

## 4. Implications for Practitioners and Researchers

The results could be useful for professors, researchers, nurses, pediatricians, and psychologists. Professors can get a summary of the topics to be included in their teaching material. Furthermore, this study can provide investigators with a complete overview of the global literature on stress, anxiety, and PPD in mothers of preterm infants in the NICU, enabling them to expand their knowledge on the subject; it can also serve as a tool to identify future research trends.

## 5. Conclusions

The aim of this study was to analyze the state of research on stress, anxiety, and PPD in mothers of preterm infants in the NICU so as to establish the cognitive and social structure of the field. Therefore, the main contribution of this manuscript was the use of bibliometrics for research in this field. On the basis of a large number of peer-reviewed articles (a total of 366 between 1976 and 2020), this research explained the productivity, cooperation, and impact indicators that help researchers to identify the main factors (authors and journals) that contribute to the development of the field.

It was also possible, on the basis of the analysis of the keywords used by the authors to characterize their research and the analysis of common occurrence, to propose certain themes to be developed in the future. Firstly, mothers of infants who require hospitalization in the NICU may experience anxiety, stress, and PPD. These emotions arise due to the NICU environment and exposure to multiple stressors, the infant’s behavior and appearance, and the infant’s role change. In addition, it can have a negative impact on the attachment between mother and child. Understanding this relationship allows early intervention to prevent the development of PPD and to strengthen mother–child interaction [4,64]. In addition, the concept of the infant is highlighted, as this condition can largely determine the life of the newborn with negative aspects for the infant [67].

Second, presenting a well-established and comprehensive bibliometric analysis can help young and established researchers expand their knowledge of stress, anxiety, and PPD in mothers of preterm infants in the NICU and, more importantly, identify gaps in the literature. The origin, evolution, and current status of the relationship linking stress, anxiety, and PPD in mothers of preterm infants in the NICU were presented, as well as the distribution of research across thematic areas, journals, and authors. Lastly, researchers working on this topic may find this research relevant as it provides information on the state of the art of current research and, thus, helps to identify future research opportunities. There is a growing body of literature on postpartum depression, anxiety, and maternal stress in neonatal intensive care units. Most of the studies included in our review focused on mothers, but some researchers included fathers, highlighting the increasing importance of involving both parents in assessing the impact of neonatal admission. A range of interventions tailored to individual needs is likely to be required to address emerging mental health symptoms during and after neonatal admission to the NICU.

This study presents some limitations. First of all, it is a characteristic feature of bibliometric studies that they are mainly founded on quantitative analysis, and it is true that, although some authors publish few papers, they have considerable impact on their field of research. Secondly, the choice of a database, Scopus, instead of another, such as the WoS database, may also be a constraint. However, Scopus was chosen because almost 84% of WoS articles can be found in Scopus [40]. Furthermore, fewer journals are indexed on WoS than on Scopus. Therefore, choosing the Scopus database diminished the risk of ignoring relevant manuscripts. Thirdly, we used the total number of citations as a measurement of impact; however, the age of the article can affect this metric. Therefore, the list of the most frequently quoted items may be dominated by several older items. Fourthly, the periods chosen were based on the decision of the researchers, although this study was built on prior articles that used SciMAT in similar studies [41,42]. Lastly, it is worth noting that this study encompassed only documents published up to 2020.

## Figures and Tables

**Figure 1 children-08-00730-f001:**
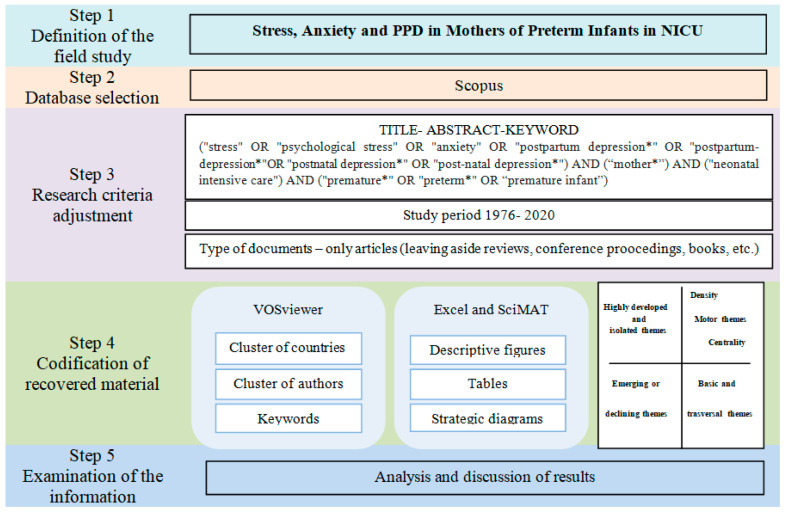
Steps of the bibliometric analysis. Source: own elaboration.

**Figure 2 children-08-00730-f002:**
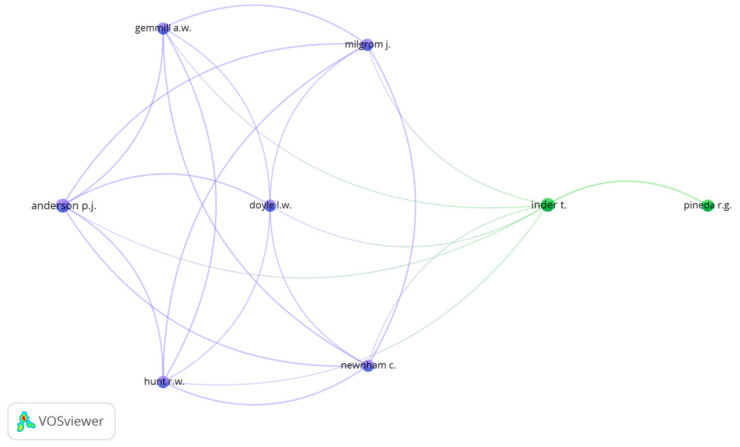
Network of relationships among authors (minimum three documents in common). Source: data from Scopus (2020), generated using VOSviewer.

**Figure 3 children-08-00730-f003:**
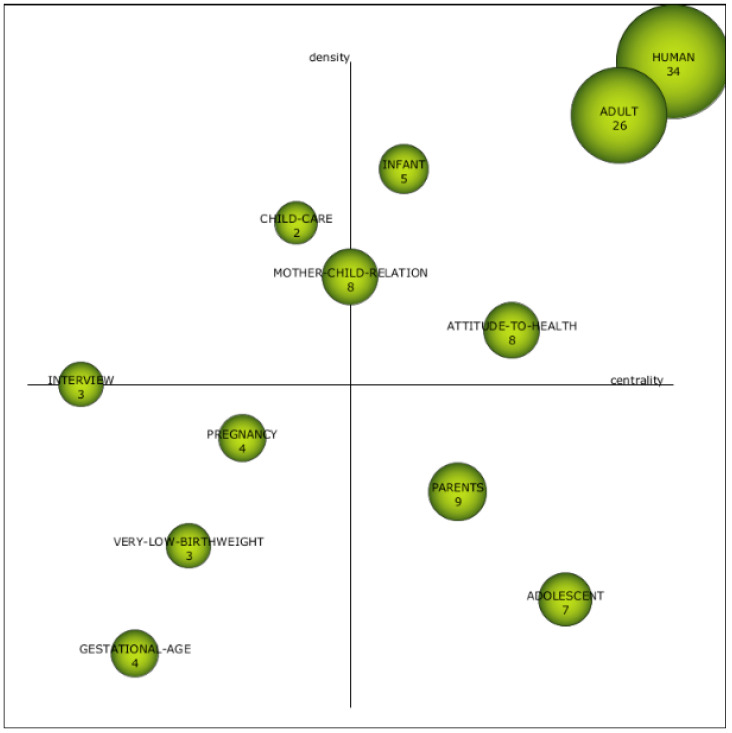
Strategic diagram from 1976 to 2002. Source: data from Scopus (2020), generated using SciMAT.

**Figure 4 children-08-00730-f004:**
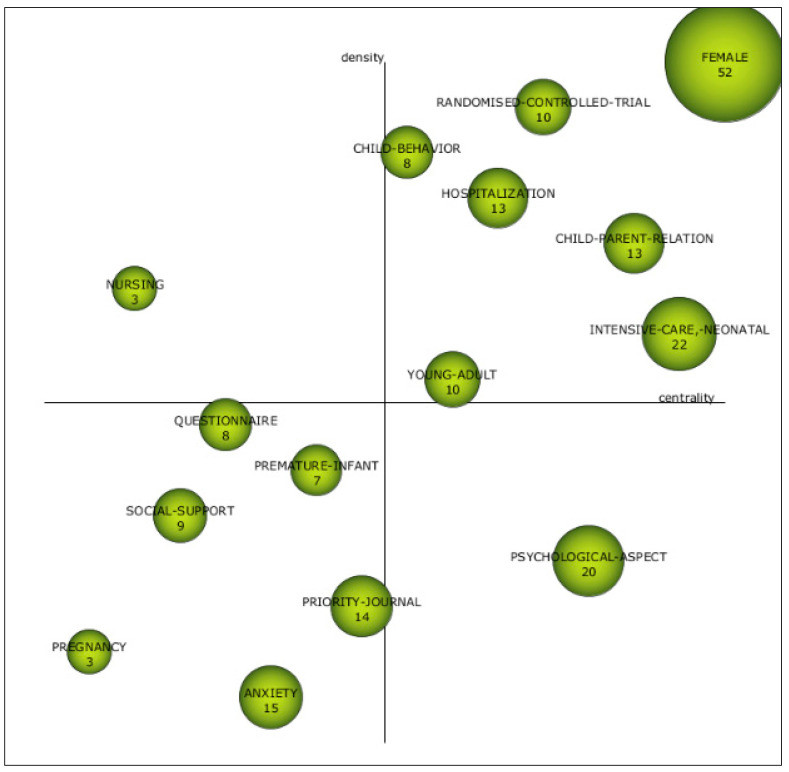
Strategic diagram from 2003 to 2011. Source: data from Scopus (2020), generated using SciMAT.

**Figure 5 children-08-00730-f005:**
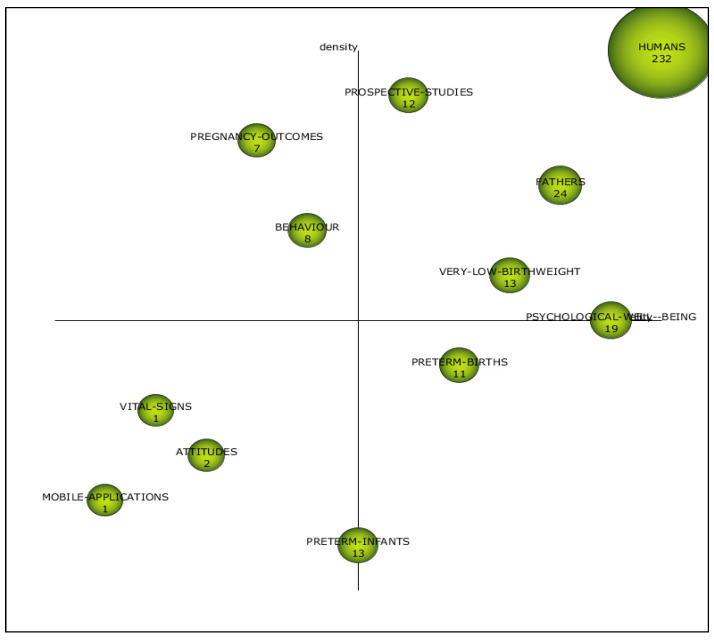
Strategic diagram from 2012 to 2020. Source: data from Scopus (2020), generated using SciMAT.

**Table 1 children-08-00730-t001:** The main characteristics of the articles on stress, anxiety and postpartum depression in mothers of preterm infants in NICU from 1976 to 2020.

Year	A	C	C/A	AU	AUA	JA	COA
2020	99	2	2	160	4.89	34	25
2019	283	5.24	1.14	160	2.96	41	28
2018	285	8.14	3.10	160	4.10	36	23
2017	255	9.81	5.81	146	5.62	24	11
2016	553	24.04	14.68	128	5.12	22	14
2015	455	22.75	17.15	104	5.20	15	15
2014	599	28.52	21.74	114	6.00	18	9
2013	913	36.52	29.08	117	4.68	21	13
2012	447	34.38	30.75	50	4.17	12	6
2011	118	19.67	13.71	31	4.43	7	7
2010	310	51.67	45.67	30	5.00	6	3
2009	584	48.67	42.42	92	7.67	16	10
2008	405	45.00	39.78	38	4.22	9	2
2007	141	28.20	25.60	21	4.20	13	2
2006	520	104.00	94.2	24	4.80	5	4
2005	344	57.33	50.67	17	2.83	6	4
2004	185	61.67	58,33	13	4.33	3	4
2003	850	141.67	133,50	25	4.17	5	3
2002	66	33.00	31,50	6	3.00	2	2
2001	60	30.00	25.5	4	2.00	2	2
2000	198	66.00	60.33	5	1.67	3	2
1999	588	294	278	7	3.50	2	2
1998	12	12.00	11	5	5.00	1	1
1997	161	53.67	50.67	7	2.33	3	3
1996	207	69.00	63.67	8	2.67	3	2
1995	379	94.75	91.25	21	5.25	3	3
1994	256	64.00	62.5	13	3.25	4	2
1993	39	19.50	18.5	7	3.50	2	2
1992	200	100.00	94.5	6	3.00	2	2
1991	13	13.00	13	1	1.00	1	1
1990	20	10.00	9.5	7	3.50	2	2
1989	28	28.00	26	4	4.00	1	1
1986	387	193.50	185	8	4.00	2	1
1984	0	0.00	0	3	3.00	1	1
1983	3	3.00	3	1	1.00	1	1
1981	58	58.00	57	3	3.00	1	1
1980	5	5.00	5	3	3.00	1	1
1976	92	92.00	92	3	3.00	1	1

A: article production per year; C: citation number per year; C/A: average citation number per article (citation total since 1976/total of articles since 1976); AU: author number per year; AUA: number of authors that published at least one article in a specific year; JA: number of journals that published at least one article in a specific year; COA: number of countries that published at least one article in a specific year. Source: own elaboration based on Scopus 2020.

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
