# Peer review of "Mapping the Field in Stress, Anxiety, and Postpartum Depression in Mothers of Preterm Infants in Neonatal Intensive Care"

_children, 2021, doi:10.3390/children8090730_

Round 1
Reviewer 1 Report
This is an informative attempt at describing the state of research on stress, anxiety and postpartum depression in mothers of preterm infants in the Neonatal Intensive Care Unit. The manuscript may be improved with attention to a few specific details:
1. Why was this search only carried out on one platform/database? The authors say “84% of manuscripts in WoS are also indexed in Scopus and 159 that Scopus contains more journals than WoS”. If this analysis is to be done in a systematic way, similar to systematic reviews or meta-analyses, it would be worth looking at all more than one (and potentially even more than two) databases including PubMed, Google Scholar, Embase etc. Of course, there will be duplicates but it is those can be removed by scanning titles. The danger in not doing so lies in potentially missing important research.
2. I know these searches take time but this paper will be out of date by the time it goes into print, especially given the findings of this manuscript that more research on this topic is being published than ever before and the aim of wanting to provide “information on the state of the art of current research”. The authors should update their search to at least include 2020, if not 2021.
3. Are those all of the search terms? What about perinatal depression? And was the asterisk used behind all of the depression terms or only after “postnatal depression*”?
4. In the Interpretation of the Analysis section there is a word missing at the end of the following sentence: “Secondly, they agree with this study, which states that the United States is the most productive geographical.”
5. Correct “infat” in Conclusion section and several double spaces between words.
6. What are the benefits of doing a bibliometric analysis over a systematic review. Could the authors please comment on that?
Author Response
Dear reviewer, thank you very much for reviewing our manuscript. We greatly appreciate the time and effort that you take for your suggestions, which we are sure, will greatly enhance our research. We also fully thank you for your positive comments. Please find in the file attached a point-by-point response to your concerns. We hope that you find our responses and changes satisfactory, and that the manuscript is now acceptable for publication.

Reviewer 2 Report
The article presented to me for review is very interesting. The articles in the topic were thoroughly analyzed. I have a few comments:
line 32: The authors cite the 2016 WHO report. There should be a reference to the literature at the end of this sentence, or after the report is cited.
line 41: as above
lines 157-158: as above
lines 130-133: please put this information in new section: Implications for practitioners and researchers - before the conclusions.
Were only full-text manuscripts considered? were the abstracts themselves taken into account? Has the language of the manuscripts been specified?
Throughout the manuscript, the citation should be corrected, the surnames of the authors are not in parentheses, and the reference number to the references, eg [1], [1,2] or [3-7]. Additionally, the list of references should be numbered according to the order in which they are cited.
Author Response
Dear reviewer, thank you very much for reviewing our manuscript. We greatly appreciate the time and effort that you take for your suggestions, which we are sure, will greatly enhance our research. We also fully thank you for your positive comments. Please find attached a point-by-point response to your concerns. We hope that you find our responses and changes satisfactory, and that the manuscript is now acceptable for publication.
COMMENT 1.line 32: The authors cite the 2016 WHO report. There should be a reference to the literature at the end of this sentence, or after the report is cited.
COMMENT 2. line 41: as above
COMMENT 3. lines 157-158: as above
RESPONSE TO COMMENTS 1, 2 AND 3.
Thank you for your comments. All references for literature have been modified and included accordingly.
COMMENT 4. lines 130-133: please put this information in new section: Implications for practitioners and researchers - before the conclusions.
RESPONSE TO COMMENT 4.
Thank you for your suggestion. A new Section named Implications for practitioners and researchers has been added before Conclusion Section with the old paragraph.
COMMENT 5. Were only full-text manuscripts considered? were the abstracts themselves taken into account? Has the language of the manuscripts been specified?
RESPONSE TO COMMENT 5.
The restriction for documents was “article” but not full text. What is more, abstracts are taken into consideration due to the fact that search formula encompasses Title -Abstract and Keywords. Thus, abstracts have been considered. Finally, no language restrictions were included in the selection of information.
COMMENT 6. Throughout the manuscript, the citation should be corrected, the surnames of the authors are not in parentheses, and the reference number to the references, eg [1], [1,2] or [3-7]. Additionally, the list of references should be numbered according to the order in which they are cited.
RESPONSE TO COMMENT 6
Thank you for your comment. We have proceeded to change all the citations accordingly.
We again want to thank you for your time and effort put in the revision.
